# Reducing versus Embracing Variation as Strategies for Reproducibility: The Microbiome of Laboratory Mice

**DOI:** 10.3390/ani10122415

**Published:** 2020-12-17

**Authors:** Vera M. Witjes, Annemarie Boleij, Willem Halffman

**Affiliations:** 1Institute for Science in Society, Faculty of Science, Radboud University, 6500 GL Nijmegen, The Netherlands; vera.m.witjes@gmail.com; 2Department of Pathology, Radboud Institute for Molecular Life Science (RIMLS), Radboud University Medical Center, 6500 HB Nijmegen, The Netherlands; Annemarie.Boleij@radboudumc.nl

**Keywords:** reproducibility, animal experiments, mouse microbiomes, variation

## Abstract

**Simple Summary:**

The rate at which biomedical animal experiments fail to reproduce expected results is of raising concern in the animal research community. One of the explanations of irreproducibility is that animals used in repeated experiments are not identical to the original animals: there is variation in the animals’ bodies, even if they are genetically similar. For example, there might be substantial differences between the microorganisms inhabiting the experimental groups of mice. For animal researchers, it is essential to know how to deal with this variation in animal experiment design. Both reducing and embracing variation are suggested as strategies for reproducibility. In this theoretical review, we use mouse microbiome variation as an example to analyze this ongoing discussion, drawing on both animal research and philosophy of science. This analysis helps articulate options for animal researchers to deal with variation in the future design of animal experiments, contributing to reproducibility.

**Abstract:**

Irreproducibility is a well-recognized problem in biomedical animal experimentation. Phenotypic variation in animal models is one of the many challenging causes of irreproducibility. How to deal with phenotypic variation in experimental designs is a topic of debate. Both reducing and embracing variation are highlighted as strategies for reproducibility. In this theoretical review, we use variation in mouse microbiome composition as an example to analyze this ongoing discussion, drawing on both animal research and philosophy of science. We provide a conceptual explanation of reproducibility and analyze how the microbiome affects mouse phenotypes to demonstrate that the role of the microbiome in irreproducibility can be understood in two ways: (i) the microbiome can act as a confounding factor, and (ii) the result may not be generalizable to mice harboring a different microbiome composition. We elucidate that reducing variation minimizes confounding, whereas embracing variation ensures generalizability. These contrasting strategies make dealing with variation in experimental designs extremely complex. Here, we conclude that the most effective strategy depends on the specific research aim and question. The field of biomedical animal experimentation is too broad to identify a single optimal strategy. Thus, dealing with variation should be considered on a case-by-case basis, and awareness amongst researchers is essential.

## 1. Introduction

In recent years, concerns over irreproducible research findings have spread across different scientific disciplines and into public debate [1], prompting the term ‘reproducibility crisis’ [2,3]. The failure to reproduce study results is also noted in the field of biomedical animal experiments, in which animals (predominantly mice) are used as models to study human disease development and test treatments [4,5,6]. An analysis by Freedman et al. revealed that more than 50% of preclinical animal studies were not reproducible [7]. 

Such evidence hampers the credibility of animal research, making it harder for researchers to rely on the work of others. Inconclusive or non-credible animal studies imply research waste, hamper translation to human medicine, and involve ethical implications. Research waste is not just a matter of squandered money and human effort, but also an abuse of test animals, breaching Russell and Burch’s principles of Reduction, Replacement and Refinement [8]. Irreproducibility complicates the search for fundamental insights and medical innovation, as well as the exploration of non-animal alternatives. 

Researchers have proposed numerous causes of irreproducibility in animal experiments, including poor experimental design, non-transparent reporting of methods, and publication bias [9,10]. Tighter research protocols and improved publication practices can address such sources of variation. However, unaccounted phenotypic variation between animals when designing experiments also contributes to irreproducibility [11]. Dealing with this source of variation is complicated and a topic of debate [12,13,14]. The question of whether to reduce or embrace phenotypic variation is crucial for the future design of animal experiments, ranging from technical concerns in animal breeding to deeper assumptions of how animal experiments render trustworthy knowledge.

One of the sources of phenotypic variation between animals is variation in host-associated microorganisms, collectively referred to as the microbiome [15,16]. In this theoretical review, we use variation in mouse microbiome composition as an example to analyze the ongoing discussion on reducing versus embracing variation in animal research. To get to this analysis, we elaborate how variation in mouse microbiome composition can affect phenotypic results and can contribute to irreproducibility. However, since the role of variation in irreproducibility touches upon fundamental conceptual issues, a theoretical clarification is required. As such, we start with a conceptual framework to define reproducibility and analyze precisely the reasons behind the research strategy of reducing variation.

## 2. Conceptual Framework: Defining and Understanding Reproducibility

‘Reproducibility’ requires further specification and clarification, as it is a broad concept which can be interpreted in different ways in different fields of research. Similar to Dirnagl [17] and Macleod [18], we use Goodman’s conceptual framework written to specify reproducibility [19]. Goodman distinguishes three forms of reproducibility. Firstly, inferential reproducibility, which refers to the ability to reach similar conclusions from the same analysis. Secondly, methods reproducibility, which refers to the provision of experimental procedures (e.g., in a publication) in sufficient detail to allow another expert to repeat the experiment, at least in principle. Thirdly, results reproducibility, which involves a new independent experiment (also referred to as a replication study) verifying the results and data [19]. 

While recognizing the relevance of other interpretations of reproducibility, this review focuses on what Goodman defines as results reproducibility. Independent replication studies that fail to confirm the original result contribute significantly to the rising concern about irreproducibility amongst the animal research community [7,18]. Dirnagl explains that a replication study can be direct, using methods and circumstances as identical as possible, or conceptual, consciously using different methods that are designed to tap into the same phenomenon as the original [17]. Macleod refers to this conceptual replication as a test of robustness [18]. 

Besides different interpretations and terminology, the epistemic function of reproducibility varies, as was described by philosopher of science Leonelli [20]. Reproducibility is often considered a foundational feature of good science: the verification of research results is crucial for both researchers and research users, as they wish to know the likelihood of the findings being true. Irreproducibility logically leads to concerns about the truth of scientific claims. However, Leonelli explains that a narrow interpretation of reproducibility as demarcating good science from bad science might be inaccurate and misleading [20]. Failure to replicate an experiment may reveal hidden factors, if the failure triggers an investigation of why replication failed. Irreproducible experiments may thus lead to more, rather than less, knowledge. Inversely, direct, identical replications may produce the same results, but may also simply repeat the previous error or conceptual bias [20].

Because of the ambivalent nature and complex causes of irreproducibility, it is important to specify its meaning. On the one hand, irreproducibility can indicate that the original study or the replication study was a false positive. False positive results can occur by chance or because measures to reduce the risk of bias were not sufficiently taken and the internal validity of the study was low [18,21]. Internal validity refers to the rigor of the study design and analysis to isolate cause and effect [22]. On the other hand, irreproducibility can occur because the result was valid only in those specific experimental circumstances. The result lacks external validity, which is defined as the extent to which the study results are generalizable to other populations or other conditions [22]. Thus, irreproducibility can be the consequence of a biasing factor in the original or replication study, or it can indicate a lack of generalizability.

Even direct replication studies, replicating the original study as identically as possible, might obtain divergent results due to the influence of as-of-yet unknown, unreported, or seemingly unimportant factors that lack generalizability. In this case, the direct replication study is not a valid one due to violation of what Meehl introduced as the ceteris paribus clause: all things else being equal—i.e., all other relevant factors remaining identical in both the original and replication study [23,24].

In animal research, direct replication studies are a challenge due to biological phenotypic variation. Questions arise as to when replication studies are considered to be direct and if they can ever be truly direct. In spite of genetic standardization strategies, there is a tremendous number of factors influencing this phenotypic variation, making reproducible research a challenge. Besides complicating direct replication studies, phenotypic variation between animals and sources of phenotypic variation is also a challenge in the design of animal experiments. In experimental designs, researchers should aim to avoid false positive results (taking into account all factors biasing the result), but also aim to ensure generalizability. Optimizing internal validity and optimizing external validity might require different approaches in dealing with biological variation, as we can illustrate with the contribution of the microbiome to phenotypic variation. 

## 3. Phenotypic Variation: Genotype-Environment/Microbiome 

Phenotypic variation in animals is a natural phenomenon crucial for population survival. Not surprisingly, phenotypic variation is also present in current stocks of laboratory mice. A prominent view in the field of mouse breeding is that the observed phenotypic variation between mice can be explained by a combination of components: variation caused by differences in genetics, variation caused by the environment, and interactions between genotype and environment [25,26]. In addition, an increasingly recognized source of phenotypic variation is the mouse microbiome [27,28]. 

The mouse microbiome is the complex community of microorganisms (including bacteria, viruses, archaea, fungi and meiofauna) and their genetic material present throughout the mouse body [29]. These trillions of microbes inhabiting all mammalian bodies have been brought into focus in biomedical research over the last few decades. Research has been able to disentangle some of the complexity of these microbial communities due to new ‘omic’ methods (i.e., genomics, metagenomics [30,31]), and has shown their relevance in human and animal health [32]. Studies have associated the microbiome with a wide range of diseases. For example, systematic reviews indicate a link between microbiome composition and colorectal cancer [33], autism [34] and type 1 diabetes [35]. The relationship between microbes and hosts consists of symbiotic, commensal, or pathogenic interactions. Thereby, microbes can influence many (if not all) aspects of host physiology, including metabolism and the immune system [36,37]. 

There is enormous diversity and complexity in the microbiome composition of mice, which is influenced by factors including (but not limited to) genetics, diet, cage, bedding material, temperature, circadian rhythms, and by horizontal transmission from the mother, littermates or mice living in the same cage [38,39,40]. In addition to these and as-of-yet unknown factors, interactions between microbes and stochastic processes are critical in shaping microbial community composition [41,42]. Together, these factors determine an immense variability in the microbiome compositions of mice: variation between mice from different research institutes or vendors, but also variation between mice within a single research institute [43,44].

Variation in the mouse microbiome influences host phenotype, but it remains unclear how this fits into the traditional phenotype-genotype-environment model [45,46]. Here, we assume the mouse phenotype is influenced by mouse genome, the environment and the microbiome as separate components, and most importantly by complex interactions between these components [47]. This is illustrated in Figure 1. These interactions can explain why mice might respond completely differently to an intervention in one study compared to another.

## 4. The Microbiome Explaining Irreproducibility of Results

Sources of variation and their interactions are a challenge in the design of mouse experiments as they might bias the result. To optimize internal validity and to facilitate reproducibility, phenotypic variation is typically minimized in mouse studies. Experiments are performed in a controlled and standardized environment using inbred mouse strains [48,49], reducing (but not removing) genetic variation. However, in study designs, the microbiome is often not taken into account as a potential source of variation, potentially thwarting this standardization strategy. As such, mouse microbiome variation is seen as a problem, and it has been associated with concerns regarding irreproducibility of results.

Already in 2016, Stappenbeck and Virgin expressed their view that the irreproducibility of studies using mice with a genetically identical background might be ascribed to variation in the host-associated microorganisms [16]. The belief that microbes influence the reproducibility of study results received more attention because of an empirical study in *Nature Microbiology*. This study revealed that the variation in susceptibility to *Salmonella* infection of genetically similar mice purchased from different vendors could be attributed to variation in their microbiomes [28]. Subsequently, an article appeared in *The Scientist* with the alarming title “Microbes May Take Some of the Blame for the Reproducibility Crisis” [50]. 

These publications raise questions about how variation in the microbiome explains the irreproducibility of results, whether this effect is always a bad thing, and about its implications for the future of animal experiments. To start with, without knowing or specifying the extent of negative effects, the microbiome can explain irreproducibility in two ways: (i) either it can act as a confounding factor causing an invalid (false positive) result in the original or replication study, or (ii) it can act as an unknown factor violating the ceteris paribus clause in direct replication studies, and the result is not generalizable.

### 4.1. The Microbiome as a Confounding Factor

The inter-individual variation in the microbiome composition of mice can be a source of bias in experimental designs. In much of biomedical experimental research, researchers try to identify a cause and effect between two variables. In mouse experiments, this usually involves studying the effect of an independent variable on a phenotypic trait of mice, using one (or more) experimental group(s) and a control group. However, the phenotypic difference between the groups of mice might be caused by variation in the microbiome instead of the independent variable under study, or by complex interactions between the microbiome and the study variable. To illustrate this confounding effect, Stappenbeck and Virgin describe problems arising from breeding one group of mice (e.g., mutants) at the research institute and purchasing the wild-type control group from a vendor outside the institute [16]. This is bad practice for several reasons, but specifically the difference in microbiome composition resulting from separate breeding puts the comparison between two groups at risk of bias [16], as is illustrated in Figure 2.

Another potential biasing effect results from the fact that microbiome composition synchronizes between mice housed in the same cage (most likely due to coprophagy) [51]. Due to this synchronizing effect, microbiome-associated phenotypes can be horizontally transferred between cohoused mice. This cage-effect can bias comparisons between groups housed in mixed cages, but also comparisons between groups housed in separate cages [42,51]. For example, when studying the effect of treatment versus placebo in two groups of mice housed in separate cages, the observed phenotypic difference between the groups of mice most logically results from the treatment (variable of study). Yet, a phenotypic effect could also be caused by differences in microbial community composition due to the aforementioned cage-effect. In addition, there might be an interaction between microbiome and treatment causing a confounding effect.

The practice of microbiome effects on experimental results may well be more complicated than the examples described here. Whether variation in microbiome composition between mice causes bias in a study depends on the research question, study design and variables of interest. However, the possibility of microbiome effects should not be neglected. If variation in microbiome composition is not taken into account as source of biological variation in experimental designs, it could lead to the overestimation of effects, the underestimation of effects, false attribution of effects to the independent variable, and methodological confounding due to complex interactions of mouse microbiomes with environment and genotype. Consequently, the results of subsequent studies that do or do not consider microbiome effects might vary widely, contributing to irreproducibility.

### 4.2. The Result Is not Generalizable to Mice Harboring a Different Microbiome 

Variation in mouse microbiomes can also explain failed replication studies without necessarily indicating a false positive or a false negative result. It could be the case that the original study result is true, but only in the exact experimental conditions of that specific study; i.e., with the same microbiomes. This lack of external validity might come to light in a robustness test comparing mice harboring different microbiomes. In addition, the microbiome might also act as an unknown factor in direct replication studies violating the ceteris paribus clause, explaining why results are not in line. Namely, because of the influence of the microbiome on mouse phenotype and the high variation in microbiome composition between mice from different facilities, the direct replication study might be testing a different phenotype. This is illustrated in Figure 3. 

Confusingly, inbred genetically similar mice (e.g., C57BL/6 mice) that can be purchased from different vendors create the illusion for researchers that they are testing similar phenotypes in replication studies. However, due to microbiome (and environmental) effects, phenotypes between mice from different vendors might vary substantially. Ivanov et al. (2008) first observed phenotypic differences between C57BL/6 mice from different sources [52]. They observed that mice from the Jackson Laboratory had remarkably low numbers of Th17 cells in comparison to mice from other sources, and this could be reversed by fecal transplantation. They revealed the difference was caused by the presence of the bacterial genera *Cytophaga*, *Flavobacter* and *Bacteroidetes* [52]. Subsequently, more studies showed phenotypic differences in mouse strains from similar genotypes from different vendors caused by variations in microbiome composition [53], with the most recent example published in 2019, observing different susceptibilities to *Salmonella* infection between mice purchased from four different vendors. They revealed this variation was caused by differences in the presence of endogenous *Enterobacteriaceae*, which presence was protective due to competition for resources [28]. 

This variation in phenotype between genetically similar mice from different vendors can explain why replication studies fail to confirm the original result. Study results might be only true for mice harboring a specific community of microbes or specific key pathogens. The difference between the original study and a replication study can be explained by the microbiome acting as the unknown factor. For example, the differences in microbiome composition between vendors can cause differences in immune responses, as was shown by Ivanov et al., which might cause variation in a response to a potential treatment. In addition, because of microbiome-drug interactions, microbiome composition can affect the bioavailability, efficacy and toxicity of drugs [54], which can also explain differences in treatment response between replication studies. Variability in microbiome composition can therefore affect replications across different research institutes, who purchase mice from different vendors.

However, less obviously, variation in microbiome composition can also explain why researchers fail to replicate their own experiment, in the same facility, with the same environmental conditions, and when purchasing similar mice from the same vendor. In a series of experiments investigating the effects of a drug on bone density in mice [55], physiologist McCabe and her team failed to obtain direct replication. One study showed a reduction in bone density, the other showed an increase and the third one showed no effect. While the exact mechanisms remained unclear, the team identified different microbiome compositions between the groups of mice and drug microbiome interactions as the most probable explanation for variations in drug response [55]. Thus, variation in microbiome composition and its influence on host phenotype can explain why some replication studies fail. 

## 5. The Microbiome and ‘Failed’ Replications: What Does It Mean?

Variation in mouse microbiome composition can explain the irreproducibility of results, but the contribution of microbiome variation to the ‘reproducibility crisis’ remains unclear. On the one hand, inadequate experimental design allowing the microbiome to act as a confounding factor could be interpreted as ‘bad science’. In contrast, if mouse microbiome variation contributes to irreproducibility by violating the ceteris paribus clause in direct replication studies, this does not automatically indicate false results or bad scientific practice. Such studies can be seen as a test of robustness. In fact, the likelihood of replication studies testing a different phenotype is extremely high, due to phenotypic plasticity and the many variables affecting mouse phenotype (of which the microbiome is only one) [21]. This even raises the question of whether animal replication studies can ever truly be direct replications, or whether they are always unintended tests of robustness [18]. 

‘Failed’ replications caused by unintended differences in latent variables, such as variation in mouse microbiome composition, are therefore not necessarily a bad thing, but could contribute to new knowledge as the hidden source of variation is investigated. For example, if a new chemotherapeutic drug for colorectal cancer is effective in a preclinical mouse study, but non-effective in a replication study, one could investigate in a subsequent study whether microbiome differences are responsible for influencing drug effectiveness. This approach, which was introduced as ‘scoping reproducibility’ by Leonelli [20], could lead to relevant new fundamental or clinical knowledge. Without ‘failed’ replications and unexplained observed variation preceding the study of Velazquez et al., we would not know that endogenous *Enterobacteriaceae* cause variation in susceptibility to *Salmonella* infection [28]. It is currently not feasible to routinely screen the microbiome composition of all mice in all experiments, and therefore this scoping reproducibility approach remains limited to studies in which researchers expect a potential microbiome effect. In contrast to traditional animal experiments, scoping reproducibility relies on trial-and-error, which can be seen as contrary to the Three R principle described by Russell and Burch, and might be disapproved of by ethics committees and scientists themselves [8]. 

A narrowly negative interpretation of irreproducibility might be inadequate. Ideally, the research community wishes studies to be reproducible, ultimately leading to reproducible treatments. Our microbiome analysis indicates that researchers should avoid bias in their study design, but also ensure that results are generalizable in the face of such hidden sources of phenotypic variation. These complications ask for different approaches in dealing with variations in experiments.

## 6. Improving Internal Validity

To avoid bias, researchers should avoid a systematic difference in microbiome composition between the experimental group(s) and control group. In general, the most well-known measures to avoid a systemic difference between two groups of mice are randomization and standardization [56]. To avoid microbiome differences, the random allocation to groups alone is not effective because of cage-effects. Additionally, when the variable under study (a gene) defines the two groups (knock-out versus control), random allocation is not applicable. This asks for a different approach. Researchers can minimize the possibility of the microbiome confounding study results by standardization within experiments using littermate controls or by cohousing groups, or by using standardized microbial consortia which allow for standardization within and across experiments.

### 6.1. Cohousing and Littermate Controls: Standardization within Experiments

The most well-established and easy method to control for microbiome effects is cohousing at weaning. Individual mice from experimental and control groups are housed together in a single cage. This should eliminate the microbiota cage-effect [42,57]. To ensure more homogenization across cages, the transfer of bedding material could supplement cohousing. Another well-acknowledged method to control for microbiome effects is designing littermate-controlled experiments. This is particularly useful when testing the effect of a gene knock-out (or mutant) versus control. The breeding between knock-out and control mice leads to heterozygous offspring in the first generation, and homozygous littermates in the second generation. Comparison between these knock-out and their control littermates controls for the effects of the microbiome [16]. Both methods are shown in Figure 4. In 2019, a study compared the two methods and concluded that littermate-controlled experiments were more effective in standardizing microbiome composition across groups compared to experiments in which mice were cohoused [58]. 

Both methods are known to be used in mouse studies, but it is impossible to estimate how often they are used and how often they should be used. Noticeably, after the plea for littermate-controlled experiments as the gold standard by Stappenbeck and Virgin [16], more studies were published using this design [59,60]. One study replicated previous studies reporting a protective role for NLRP6 against colitis via regulation of healthy microbiota. However, in contrast to previous studies, the replication study used littermate controls and revealed that NLRP6 did not impact gut microbiota and colitis control. Previous studies might have been biased by a pro-colitogenic microbiota composition in the control mice at baseline [60]. This example shows how important it can be to avoid the microbiome as a source of bias, and encourages researchers to carefully consider their experimental design. 

However, despite being effective under some conditions, there are also drawbacks to the use of littermate controls and cohousing groups. A disadvantage to the use of littermate controls is the production of excess mice (heterozygous), which contrasts the principle of reduction and might not be approved by animal ethics committees or scientists themselves. Additionally, littermate controls involve additional work because researchers need to genotype the second-generation mice before the experiment. A downside to cohousing is the limit to the number of mice per cage, and for practical reasons it is not always possible to cohouse experimental and control groups. Therefore, to minimize bias, some researchers propose or consider standardizing microbiome composition between mice not only within but also across experiments [61,62]. 

### 6.2. Defined Microbial Consortia: Standardization within and across Experiments

There have been several attempts to standardize the microbiome within mouse strains. Selective inbreeding was proven to be ineffective in harmonizing gut microbiome composition [63]. Alternatively, MacPherson and McCoy have suggested colonizing germ-free mice with a defined microbial consortium, and thereby creating ‘isobiotic’ strains of mice [61]. To ensure reproducibility, isobiotic strains should have a stable microbiome composition across multiple generations, and mice should be fed a sterile diet the ingredients of which are openly available. Additionally, the microbial consortium should be transferable to germ-free mice from a different genotype or different institution [61,64]. 

Due to the variability in microbiome composition, it is extremely difficult to meet these criteria, but over the years, efforts have been made to develop stable standardized consortia. The most well-known defined microbial consortium is the Altered Schaedler Flora, which was developed in 1987 [65]. Subsequently, researchers have developed a more diverse and more easily accessible microbial consortium, namely the stable Defined Moderately Diverse mouse Microbiota 2 (sDMDMm2) [66]. This consortium was relatively stable (if mice were fed a standardized diet) over time and transfer to other institutes was successful [64]. To date, these consortia are effectively used to study host-microbiome interactions [67,68], but to our knowledge are not generally used to avoid bias in animal experiments.

MacPherson and McCoy suggest that more effort should be put into creating standardized microbial consortia meeting proposed criteria [61]. They argue that the targeted use of isobiotic mouse models would increase the internal validity of studies, ensure reproducibility across different institutes, and allow for smaller sample sizes. Thereby, they argue this approach would contribute to the Three R principle by reducing the number of mice [61,64]. However, the question remains whether such results would be generalizable to other populations, or whether isobiotic mice would favor a certain (clinical) population.

## 7. Are Animal Experiments Over-Standardized? 

The argument used by MacPherson and McCoy is in favor of standardizing mouse microbiome composition, and represents precisely the reason why scientists have introduced standardization in animal research historically. Standardization was introduced to avoid confounding factors (improve internal validity), to reduce the number of animals, and to allow for better comparison between different research institutes [69]. Animal experiments have become increasingly standardized over the years. Genetic and environmental standardization to get rid of phenotypic variation created more homogeneous study populations. As microbiome composition is highly influenced by environmental and genetic factors, this standardization might be beneficial. However, a more recent move is arguing against the traditional belief that standardization ensures reproducible results. 

### 7.1. The Standardization ‘Fallacy’ 

Experimental standardization increases the risk that the study result is true only under specific experimental circumstances, leading to idiosyncratic results [12,21]. We have illustrated that variation in mouse microbiome composition can explain irreproducibility when results lack external validity (e.g., the result is true only for mice harboring a specific microbiome). Besides variation in microbiome composition, there are unavoidable differences between animal facilities (e.g., different animal caretakers, different animal handling, different perceptions of ‘good’ research practice) [70,71,72]. Therefore, to ensure robust reproducibility, study results must be generalizable in spite of such variation. Standardization aims to improve reproducibility by removing this variation. As variation occurs in natural populations, this standardization may come at the expense of external validity, which is introduced as the ‘standardization fallacy’ [73].

Voelkl et al. therefore advocate a paradigm shift to focus on including systematic heterogenization in animal experiments [14]. Known sources of variation should be included in the study design, either as random effects or as fixed effects. They provide a wide overview of possibilities regarding how to include heterogenized study populations, and have demonstrated the effectiveness of their approach [14,74]. The perspective of embracing variation is also supported by a recent meta-analysis of variability in animal models of ischaemic stroke [75].

In light of the variation in microbiome composition, one could deliberately include mice obtained from different vendors known to harbor a varied microbiome composition, or design a multi-laboratory study. If the study result is true for a heterogenized study population, it is more likely to be reproducible, and eventually translatable to human medicine.

### 7.2. Standardized Clean Laboratories to Model the Dirty World

Another argument against standardization is that experimental conditions do not mirror the world outside the controlled experiment. In an effort to keep the environment as standardized and clean as possible, animal facilities might have accidently eliminated the complexity of mouse microbiomes affecting reproducibility and translational value. When detected in a facility, infectious pathogens are eliminated to avoid biasing of results. These clean laboratories do not mirror the natural ‘dirty’ environmental circumstances, and hence laboratory mice cannot model the physiology of wild animals (including humans) due to a less diverse microbiome composition in sanitized environments [76,77]. 

Over the last few years, efforts have been made to improve the translational value of mouse experiments by modeling a more diverse microbiome composition and natural environment. Rosshart and colleagues studied engrafting natural microbiota from wild mice into laboratory mice. Their study revealed that mice with natural (wild) microbiota had lower rates of inflammation and were more resistant against inflammation-induced tumorigenesis [76]. A more recent study investigated transferring C57BL/6 laboratory mice embryos into female wild mice. This new approach resulted in a colony of mice which they referred to as ‘wildlings’ [78]. Wildlings resembled wild mice in their microbiota (at different body sites) and their immunological profile in blood and spleen. In contrast to the microbiome of conventional mice, the wildlings’ microbiome was stable across generations and more stable to environmental changes. This, according to the researchers, could enhance the validity of research results and improve reproducibility across research institutes [78].

Thus, standardization and clean environments might have resulted in mouse models in which human disease recapitulation is hampered. In addition, where standardization and sanitation aim to reduce differences between laboratories, they might have led to increasing differences in microbiome composition. Namely, in comparison to wild mouse microbiota, the microbiota of conventional laboratory mice are less resilient to environmental changes [78]. Subtle changes in environment have more impact on microbiome composition in a standardized laboratory environment in comparison to a natural environment. Therefore, standardization might have complicated reproducibility by creating mouse models with unstable (less resilient) microbiomes.

## 8. The Future: Reducing Versus Embracing Variation?

In this review, we have illustrated that variation in mouse microbiome composition can explain the irreproducibility of results in two ways: (i) the microbiome acts as a confounding factor in study designs, and (ii) the original result was not generalizable to mice harboring a different microbiome. Paradoxically, avoiding confounding factor (i) asks for increased standardization and reducing variation, whereas (ii) ensuring generalizability asks for increasing external validity and embracing variation in study designs. This demonstrates the complications of dealing with variation in the future design of animal studies.

Reducing and embracing variation are contrasting approaches, aiming at a different research strategy. By embracing variation, increasing external validity, one aims to ensure robustness and reproducibility ‘no matter what’. The underlying mechanism may remain unknown, as there may be uncontrolled factors. Reducing variation increases internal validity, aiming for reproducibility through maximum knowledge of the system under study, allowing one to correctly identify when something is a replication. Together, these two contrasting approaches demonstrate the broad interpretability of the term reproducibility. There might even be a trade-off between the two approaches: by focusing on robustness, you might lose sight of the underlying mechanisms, whereas by focusing on a standardization, you might limit your findings to the controlled system, hindering generalization. 

Applying this to the case of variation in mouse microbiome composition, we conclude that researchers should be very aware that microbes can influence study results, but it depends on the research question and aim of the study as to how to deal with this variation. For some studies, especially fundamental studies aimed at identifying disease mechanisms, increased control (standardization within experiments) is extremely important to avoid confounding. Other, more translational studies might require robustness tests in mice harboring different microbiomes to ensure generalization. In addition, some studies might require mice harboring a ‘wilder’ natural microbiome to model the dirty world we humans live in. Dealing with variation in mouse microbiome composition is complex and should be considered on a case-by-case basis.

## 9. Conclusions

Reducing versus embracing variation as strategies for reproducibility is a central topic for the future of animal experimentation. Here, we used variation in mouse microbiome composition as an example to illustrate that dealing with variation is extremely complex. We conclude that the most effective strategy depends on the research question and aim of the specific experiment. Advocating for a single strategy would undermine the broadness of the field of biomedical animal experimentation. Careful consideration of how to deal with variation and which strategy to apply is essential for every single experiment. As such, awareness amongst researchers is most important. 

## Figures and Tables

**Figure 1 animals-10-02415-f001:**
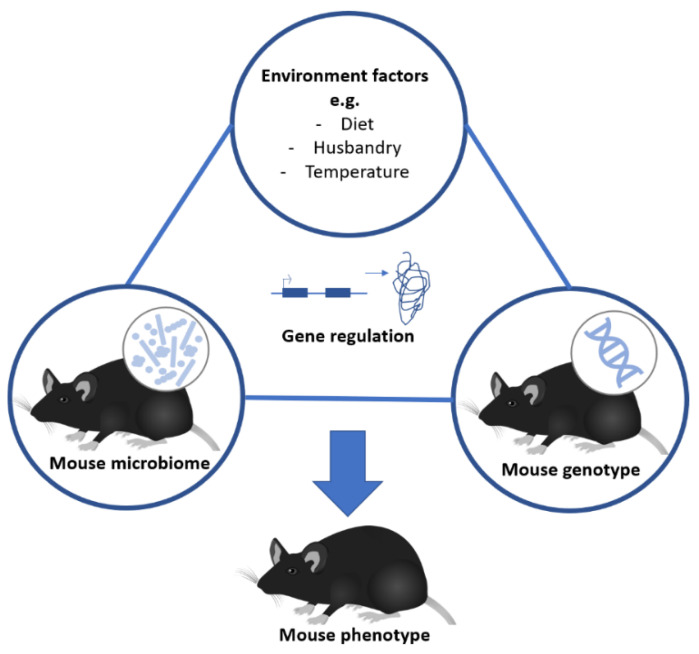
Complex interactions between environment, mouse genotype and mouse microbiome might affect mouse phenotype.

**Figure 2 animals-10-02415-f002:**
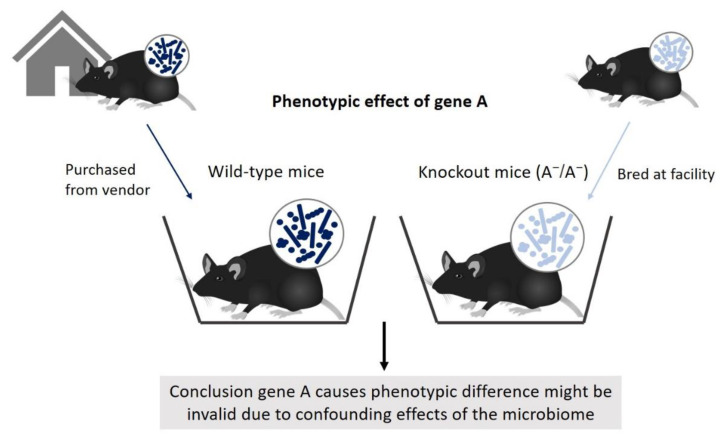
Comparisons between wild-type and knockout mice are at risk of bias due to differences in microbiome composition resulting from separate breeding.

**Figure 3 animals-10-02415-f003:**
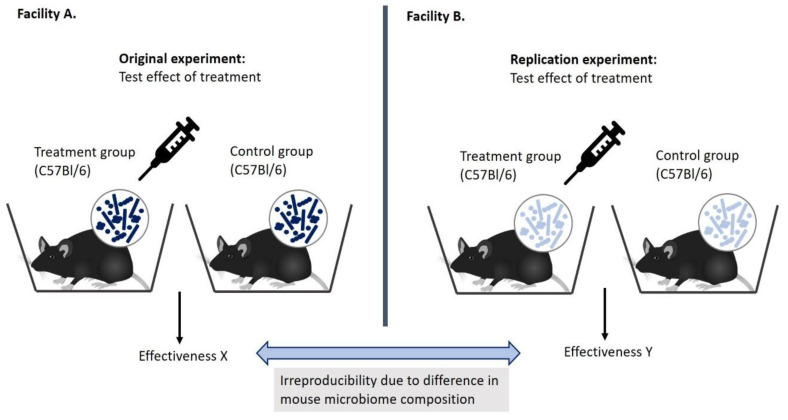
Variation in mouse microbiome composition can explain irreproducibility between (and also within) facilities as the effectiveness of a treatment might be specific to a certain microbiome composition and not generalizable to mice harboring different microbes.

**Figure 4 animals-10-02415-f004:**
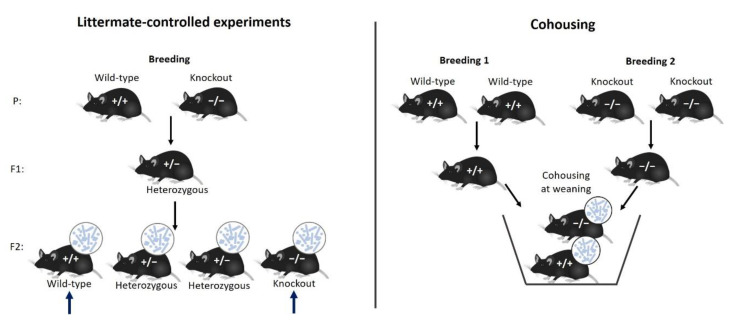
Littermate-controlled experiments and cohousing to standardize mouse microbiome composition within an experiment to avoid confounding.

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
