# Peer review of "Reducing versus Embracing Variation as Strategies for Reproducibility: The Microbiome of Laboratory Mice"

_animals, 2020, doi:10.3390/ani10122415_

Round 1

Reviewer 1 Report

The authors present an interesting discussion on the role of phenotypic variation in reproducibility of experiments using animals using the microbiome. They raise a topic that has been of great interest to the discussions around experimental rigor in raising the question of variability introducing confounding or improving generalizability, as it is capable of both.  They come to the conclusion that this should be considered, like most variables, in the design of the experiment to support the experimental aims.

This manuscript is timely and relevant in helping scientists develop awareness around this methodology and understand the use of variability to make experiments more robust.  I have a few minor comments for clarity.

Comments follow:

Lines 48-54 do an excellent job in describing the consequences of irreproducibility.  The authors may go so far to make a parallel between research waste and the erosion of lay public trust in science and support for continued use of animals which then introduces the following paragraph – the end of which 61-63 is making this point.  I think it is fair to say it is more than “now even seems crucial for…” as it is well understood that public support for research using animals relies on balanced benefit being realized.

Excellent that authors included 96-99, the potential for ‘failed’ reproducibility to give new insight.

Ln 125-131, also the assumption that inbred mice = identical genetics despite different derivation practices from the original lines.

144-150, comprehensive, yes!

276-285, nice summary of how this could be used in a favorable way – however what is the practicality of all researchers studying the microbiome, do you see this becoming similar to a routine clinical pathology lab screen (e.g. CBC/chem?)

285 – the authors need to be more clear with the concept of reduction, which is not just a simple reduction in animal numbers, but in intent this means appropriately designed and analysed animal experiments that are robust and meaningfully contribute to the scientific knowledge base.  Essentially the authors are showing how to appreciate this principle in a different way.

355 – This circles back to my question earlier, what is the practicality of using isobiotic mouse models from the authors perspective?  Does the proposed criteria for the microbiome favor a certain clinical population?

420 – this paragraph header is not consistent with it coming as a question or if this is a statement that the subsequent recommendation by the authors that it is situationally dependent.

Congratulate the authors on an excellent manuscript that will be a tremendous contribution to understanding the roll of variability towards improved translation.

Author Response

Response to Reviewer 1 Comments

The authors present an interesting discussion on the role of phenotypic variation in reproducibility of experiments using animals using the microbiome. They raise a topic that has been of great interest to the discussions around experimental rigor in raising the question of variability introducing confounding or improving generalizability, as it is capable of both.  They come to the conclusion that this should be considered, like most variables, in the design of the experiment to support the experimental aims.

This manuscript is timely and relevant in helping scientists develop awareness around this methodology and understand the use of variability to make experiments more robust.  I have a few minor comments for clarity.

Comments follow:

Point 1: Lines 48-54 do an excellent job in describing the consequences of irreproducibility.  The authors may go so far to make a parallel between research waste and the erosion of lay public trust in science and support for continued use of animals which then introduces the following paragraph – the end of which 61-63 is making this point.  I think it is fair to say it is more than “now even seems crucial for…” as it is well understood that public support for research using animals relies on balanced benefit being realized.

Response point 1: We thank you for raising this point, and we agree that saying “now even seems crucial..” is an underestimation considering the rest of our manuscript. We have revised the sentence to make it more convincing (line 61).

Point 2: Excellent that authors included 96-99, the potential for ‘failed’ reproducibility to give new insight.

Point 3: Ln 125-131, also the assumption that inbred mice = identical genetics despite different derivation practices from the original lines.

Response point 3: The assumption that inbred mice are genetically identical is indeed misleading and should be avoided in our manuscript. However, adding this information does not fit well within the proposed lines. We have now emphasized the fact that inbred mice are genetically similar but not identical in lines 164-165.

Point 4: 144-150, comprehensive, yes!

Point 5: 276-285, nice summary of how this could be used in a favorable way – however what is the practicality of all researchers studying the microbiome, do you see this becoming similar to a routine clinical pathology lab screen (e.g. CBC/chem?)

Response point 5: Thank you for explaining that the practicalities of ‘scoping reproducibility’ are not clear. We do not think it is feasible to routinely screen the microbiome of all mice in all experiments or to store fecal material of all mice in all experiments due to practical limits. In this paragraph we aim to indicate that if a replication study fails to confirm the original result, and the researchers expect a potential microbiome effect, this could be further investigated and potentially lead to new fundamental knowledge. We have added an extra sentence (lines 286- 289) to clarify this point.

Point 6: 285 – the authors need to be more clear with the concept of reduction, which is not just a simple reduction in animal numbers, but in intent this means appropriately designed and analysed animal experiments that are robust and meaningfully contribute to the scientific knowledge base.  Essentially the authors are showing how to appreciate this principle in a different way.

Response point 6: We completely agree that our manuscript shows how to appreciate the principle reduction in a different way: our analysis helps researchers to articulate options to deal with experimental variation, contributing to reproducibility and robust meaningful science. In this specific sentence (ln 289 - 291) we aim to explain that ‘scoping reproducibility’ relies on trial-and-error in contrast to traditional animal experiments, and therefore it might be seen as undermining the principle reduction. We have revised the sentence to clarify this. We do feel it is important to mention this point as authorities emphasize the reduction of animal numbers, and might not approve all strategies.

Point 7: 355 – This circles back to my question earlier, what is the practicality of using isobiotic mouse models from the authors perspective?  Does the proposed criteria for the microbiome favor a certain clinical population?

Response point 7: We wish to emphasize that this line is not the authors perspective. The practicalities of isobiotic mice (standardization) and potential downsides are explained in the paragraph ‘are animal experiments over-standardized?’. Here, we explain that irreproducibility can also be caused by a lack of generalizability. By a lack of generalizability we mean that the result is only true for a specific population, and thus indeed the ‘isobiotic’ mice could favor a certain (clinical) population. Thank you for raising this point, we have added an extra sentence to bring the two paragraphs together (line 364-366).

Point 8: 420 – this paragraph header is not consistent with it coming as a question or if this is a statement that the subsequent recommendation by the authors that it is situationally dependent.

Response point 8: We agree this paragraph header should be formulated as a question, so we have added a question mark (line 427).

Point 9: Congratulate the authors on an excellent manuscript that will be a tremendous contribution to understanding the roll of variability towards improved translation.

Response point 9: Thank you for your generous comments.

Reviewer 2 Report

To address the reduction versus inclusion of variance as strategies for reproducibility using the example of the microbiome in experimental mice is very welcome. This is an important article against the background of the reproducibility crisis and the credibility of science especially in the context of animal experiments. In my view, the influence of animal welfare legislation against the background of the EU Directive 2010/63 has not been sufficiently taken into account and should be supplemented. As the ultimate aim of the Directive is to eliminate animal testing and the 3Rs need to be fully integrated by then, the approval authorities have put a strong emphasis on reduction in animal testing applications and have therefore had a strong influence. This must still be considered in your article (e.g. on p.7 l: 251 and p.8 L, 326) heterogeneous breeding produces more surplus animals and was not approved against the background of reduction or was avoided by the scientists. Please complete this point. Therefore, I have recommended the publication after minor revision.

Author Response

We thank you for your feedback and comment. We agree that we can elaborate that authorities have put a strong emphasis on reducing the number of animals, and that they might not approve all the proposed strategies. We have added two sentences (line 291, line 332-333) to emphasize this point. 
